# Calibration of sea ice drift forecasts using random forest algorithms

Cyril Palerme[1] and Malte Müller[1]

[1]Development Centre for Weather Forecasting, Norwegian Meteorological Institute, Oslo, Norway

**Correspondence:** Cyril Palerme (cyril.palerme@met.no)

**Abstract.**

Developing accurate sea ice drift forecasts is essential to support decision making of maritime end-users operating in the Arctic. In this study, two calibration methods have been developed for improving 10-day sea ice drift forecasts from an operational sea ice prediction system (TOPAZ4). The methods are based on random forest models (supervised machine learning) which were trained using target variables either from drifting buoy or synthetic-aperture radar (SAR) observations. Depending on the calibration method, the mean absolute error is reduced, on average, between 3.3 % and 8.0 % for the direction, and between 2.5 % and 7.1 % for the speed of sea ice drift. Overall, the algorithms trained with buoy observations have the best performances when the forecasts are evaluated using drifting buoys as reference. However, there is a large spatial variability in these results, and the models trained with buoy observations have particularly poor performances for predicting the speed of sea ice drift near the Greenland and Russian coastlines compared to the models trained with SAR observations.

## 1 Introduction

Passive microwave observations of sea ice concentration have been available for more than 40 years, and have shown negative trends in Arctic sea ice extent since the beginning of the satellite era (e.g., Cavalieri and Parkinson, 2012; Comiso et al., 2017), with particularly strong trends during the summer (e.g., Comiso et al., 2017). There have been less satellite observations of sea ice thickness, and these retrievals have mainly been restricted to the winter due to issues related to surface melting during the summer (Ricker et al., 2017; Petty et al., 2020). Nevertheless, long-term negative trends in sea ice thickness have also been assessed by comparing retrievals from satellite altimeters (ICESat and CryoSat-2) with submarine measurements during the period 1958 - 2000 (Kwok and Rothrock, 2009; Kwok, 2018). Furthermore, an acceleration of sea ice drift has been observed using drifting buoys and satellite observations (Rampal et al., 2009; Spreen et al., 2011; Tandon et al., 2018; Tschudi et al., 2020), and has been suggested as being a consequence of decreases in sea ice thickness and concentration due to reduced sea ice strength (Rampal et al., 2009; Olason and Notz, 2014; Tandon et al., 2018).

As a result of these changes, the Arctic ocean is becoming more accessible to marine operations, and there is an increase in maritime traffic (Eriksen and Olsen, 2018; Berkman et al., 2020). In order to ensure maritime safety, it is essential that accurate sea ice information is delivered to marine end-users. National ice services manually produce high-resolution sea ice charts using retrievals from various satellites such as passive microwave radiometers, optical instruments, and synthetic aperture radars (SAR). In addition to sea ice charts, short-term sea ice forecasts are also necessary for planning activities and providing

up-to-date information to end-users. However, the spatial resolution of current sea ice models is often too coarse compared to user needs.

Short-term sea ice drift forecasts are operationally produced by numerical prediction systems, but are affected by biases despite the numerous efforts for improving the models (Hebert et al., 2015; Schweiger and Zhang, 2015; Rabatel et al., 2018; Williams et al., 2019). Hebert et al. (2015) evaluated sea ice drift speed forecasts from the U.S. Navy's Arctic Cap Nowcast/Forecast system. They found that the predicted ice drift speed was slower than drifting buoys in the summer months, and that a persistence forecast was generally better than the forecasts from the prediction system during the summer. In contrast, the forecasts produced by the U.S. Navy's Arctic Cap Nowcast/Forecast system outperformed persistence forecasts during the winter months. Schweiger and Zhang (2015) evaluated forecasts of sea ice drift speed from the Marginal Ice Zone Modeling and Assimilation System (MIZMAS) and found root mean square errors from 4.5 to 8 km per day for lead times of 1 and 9 days, respectively. These forecasts outperform a climatological reference for all lead times (up to 9 days). Sea ice drift forecasts from the neXtSIM-F system have been evaluated by Rabatel et al. (2018) and Williams et al. (2019), and root mean square errors of about 3 and 4 km per day have been reported for lead times of 1 and 4 days, respectively (Williams et al., 2019).

Sea ice drift is influenced by various sea ice characteristics such as concentration and thickness, as well as by near-surface wind and ocean currents (Rampal et al., 2009; Spreen et al., 2011; Olason and Notz, 2014; Yu et al., 2020). Though sea ice drift is mainly driven by the wind in areas with a low sea ice concentration, the relationships between these variables and sea ice drift are complex and not linear in most of the ice-covered areas (Yu et al., 2020). In order to improve the accuracy of sea ice drift forecasts, we have developed two calibration methods using random forest algorithms (Breiman, 2001), which is a supervised machine learning technique suitable for assessing nonlinear relationships between a set of predictors and a target variable.

While random forest methods have been widely used in sea ice remote sensing (Miao et al., 2015; Han et al., 2016; Lee et al., 2016; Gegiuc et al., 2018; Park et al., 2020), as well as in weather forecasting (Gagne et al., 2014; Ahijevych et al., 2016; Herman and Schumacher, 2018; Loken et al., 2019; Mao and Sorteberg, 2020), there has been less interest in using random forests in sea ice forecasting. Recently, Kim et al. (2020) developed and compared 1-month sea ice concentration forecasts based on random forests and convolutional neural networks. They obtained more accurate results using convolutional neural networks probably due to the larger learning capacity of convolutional neural networks compared to random forests, in particular to extract spatial features from the predictors (Kim et al., 2020). Furthermore, other machine learning and statistical methods have been used for sea ice forecasting, particularly for predicting the sea ice concentration and extent. Wang et al. (2019) used a vector autoregressive model and a vector Markov model to predict sea ice concentration at subseasonal time scales, and obtained the best results using the vector Markov model. The vector Markov model also significantly outperformed the National Centers for Environmental Prediction (NCEP) Climate Forecast System, version 2 (NCEP CFSv2) for lead times between 2 and 6 weeks. Comeau et al. (2019) used a method based on analog forecasting for predicting Arctic sea ice area and volume anomalies at seasonal time scales, and obtained improvements compared to damped persistence forecasts. Moreover, various neural networks have been used for predicting sea-ice concentration, and found to be skillful for 1 and 12 month forecasts (Chi and Kim, 2017; Kim et al., 2020), but only slightly better than persistence forecasts for short-term prediction

(Fritzner et al., 2020). Nevertheless, there has not been any attempt to calibrate short-term sea ice drift forecasts using advanced statistical methods.

The random forest models developed in this study are based on predictor variables from sea ice forecasts produced by the Copernicus Marine Environment Monitoring Service's (CMEMS) TOPAZ4 prediction system (Sakov et al., 2012), wind forecasts from the European Centre for Medium-Range Weather Forecasts (ECMWF), and sea ice satellite observations from the Ocean and Sea Ice Satellite Application Facility (OSI-SAF). While all the models use the same predictor variables, two sets of models were developed using either drifting buoy displacements or SAR observations for the target variables. The data and methods used in this study are presented in sections 2 and 3, respectively. In section 4, the daily SAR observations used for analyzing the spatial variability of the forecast errors, as well as for training some of the random forest algorithms, are evaluated using buoy observations. Then, the performances of the calibrated forecasts are evaluated and compared to those from the TOPAZ4 forecasts in section 4. The discussion and conclusion of this study are presented in section 5.

## 2  Data

### 2.1  Sea ice drift observations

In this study, satellite sea ice drift observations from the CMEMS product named SEAICE_GLO_SEAICE_L4_NRT_OBSERVATIONS_01 (MOSAIC version 2.0, hereafter referred as CMEMS SAR MOSAIC product) were used for training some random forest algorithms (see section 3), as well as for analyzing the spatial variability of the performances of sea ice drift forecasts. This product provides sea ice drift fields derived from SAR observations acquired by the Sentinel-1 satellites with a spatial resolution of 10 km and a temporal resolution of 24 hours. Only a fraction of the Arctic is covered by this product every day, and there is no observations north of 87.7°N (figure 1). It is worth noting that this product is an average of drift vectors derived from pairs of SAR images which are not necessarily acquired around midnight, and that the averaging process introduces uncertainties in the product. However, this product has the advantage of providing gridded sea ice drift fields with fixed time spans (from midnight to midnight the next day) which are necessary for training random forest algorithms. Moreover, the fixed time spans ease the comparison between SAR observations and sea ice drift forecasts with daily time steps.

In addition, data from the International Arctic Buoy Programme (IABP) were also used for training some random forest algorithms (see section 3), as well as for evaluating the SAR observations and the sea ice drift forecasts. For consistent comparisons with the SAR observations and the sea ice drift forecasts, the speed and direction of sea ice drift were calculated using the geographical coordinates of the buoys at midnight (UTC). The drift vectors from buoy observations were then projected onto the polar stereographic grid used in the TOPAZ4 system. When several buoys were located in the same grid cell, only the nearest one from the grid point was taken into account. In order to avoid inaccurate and unrealistic values, only the buoys with a speed between 0.1 and 100 km per day, located in an area with a sea ice concentration higher than 10 %, and further than 50 km from the coastlines were used for verification. While only the buoys with a speed between 0.1 and 100 km per day were used for training the random forest models predicting the direction of sea ice drift, all the buoys with a speed lower than 100 km per day were used for training the models predicting the speed of sea ice drift in order to make them able to predict very

low speed. During the period from June 2013 to May 2020, about 4.5 % and 0.1 % of the buoys had a speed lower than 0.1 km per day and higher than 100 km per day, respectively. The number of buoy observations used for evaluating the forecasts during the period from June 2020 to May 2021 varies between 19276 and 19576 depending on the lead time, and has been mapped in figure 1 b).

## 2.2 Predictor variables

The list of predictor variables is the same for all the models developed in this study, and can be divided into three different categories. First, some geographical information is used with the Cartesian coordinates of the grid points (x and y in the stereographic projection from the TOPAZ4 system) and the distance of the grid point to the nearest coastline in the TOPAZ4 system. Then, the sea ice concentration from passive microwave observations during the day preceding the forecast start date is also used as predictor variable. The variables from sea ice and wind forecasts during the predicted lead time can be

considered as the last category. These variables are the wind direction and speed from ECMWF forecasts, as well as the sea ice concentration, thickness, drift speed and direction from TOPAZ4 forecasts. Furthermore, the sea ice drift and concentration observations, as well as ECMWF wind forecasts, were projected onto the grid used in the TOPAZ4 prediction system using nearest-neighbor interpolation before developing the random forest models.

For the sea ice concentration observations during the day preceding the forecast start, the version 2 of the global sea ice

concentration climate data record from OSI-SAF (Lavergne et al., 2019) was used for training the algorithms. This dataset has a spatial resolution of 25 km and is available with a latency of 16 days. Therefore, it cannot be used for producing operational forecasts. Since June 2020, calibrated forecasts have been produced daily, and near-real-time sea ice concentration products at 25-km resolution processed at the Norwegian Meteorological Institute from AMSR2 and SSMIS DMSP-F16, F17, and F18 sensors based on the algorithms introduced in Lavergne et al. (2019) have been used. The AMSR2 observations have been used

when they were available, and have been replaced by SSMIS observations when they were missing.

TOPAZ4 is a coupled ice-ocean model for the North Atlantic and the Arctic which provides 10-day forecasts at a spatial resolution of 12.5 km, as well as a reanalysis (Sakov et al., 2012). It uses the version 2.2 of the Hybrid Coordinate Ocean Model (HYCOM; Bleck, 2002; Chassignet et al., 2006) coupled with a one thickness category sea ice model using an elastic-viscous-plastic rheology (Hunke and Dukowicz, 1997) derived from the version 4.1 of the Community Ice CodE (CICE). The model

native grid created using conformal mapping has a spatial resolution between 12 and 16 km in the whole domain. An ensemble Kalman filter is used to assimilate satellite sea ice and oceanic observations such as sea ice concentration and drift, along-track sea-level anomalies, sea-surface temperature, as well as in-situ temperature and salinity profiles. Moreover, TOPAZ4 is forced by ECMWF high-resolution weather forecasts at the ocean surface. While TOPAZ4 forecasts are produced daily, data assimilation is only performed on Thursdays, and only the forecasts starting on Thursdays are stored in the long-term archive.

Though the TOPAZ4 system provides forecasts with hourly time steps, the forecasts with daily outputs were used here due to the 24-hour span of SAR observations. Previous studies have reported that the speed of sea ice drift is overestimated in the TOPAZ4 system compared to buoy observations from the IABP (Sakov et al., 2012; Xie et al., 2017).

In addition to the OSI-SAF observations and TOPAZ4 forecasts, 10-meter wind forecasts from ECMWF are also used for the predictor variables in the random forest algorithms. These forecasts have lead times up to 10 days, and the model's spatial resolution changed from about 16 km to 9 km in March 2016 (https://www.ecmwf.int/en/forecasts/documentation-and-support/changes-ecmwf-model).

## 3 Methods

### 3.1 Development of random forest models

Random forest algorithms consist of an ensemble of decision trees used for regression or classification tasks (Breiman, 2001). In order to avoid overfitting (meaning that the models learn from noise in the training data), independent decision trees must be developed. The independence of decision trees is ensured by using different subsets of the training data set for developing each decision tree, as well as by randomly selecting a fraction of the predictor variables at each node (the node is then split using the variable maximizing a dissimilarity metric among the selected predictors). Each decision tree is trained with a data set created using the bootstrap method, which consists of randomly selecting samples from the original training data with replacement for creating a new data set of the same size as the original one. This results in using about 63 % of the samples from the original data set for training each decision tree.

In this study, random forest models were developed for regression using the Python library Scikit-learn-0.23.2 (Pedregosa et al., 2011), and the mean squared error was used to measure the quality of the splits. Different models were developed for predicting the direction and speed of sea ice drift, as well as for different lead times (1 to 10 days). Moreover, two sets of models were developed using target variables either from buoy displacements or from SAR observations. Therefore, 20 different models were developed using buoy displacements, and 20 other models were developed using SAR observations. In order to optimize some parameters of the algorithms, sensitivity tests were performed using only data from the training periods (see supplementary material). For these sensitivity tests, the random forest models were trained using data from about 80 % of the forecast start dates (randomly selected) within the training periods. Then, the data from the remaining forecast start dates were used for evaluating the forecast performances. This selection prevents using neighboring grid points with very similar conditions in the training and validation data sets, and was repeated 10 times in order to obtain robust results. Furthermore, the random forest models were evaluated using the same product as the one used for training for these sensitivity tests (CMEMS SAR MOSAIC product for those trained with SAR observations, and IABP buoys for those trained with buoy observations). This method was also used to evaluate the optimal fraction of the grid points covered by SAR observations used for training some random forest models (see section 3.2), as well as to assess the importance of the predictor variables (see section 3.5). Based on the sensitivity tests, we decided to develop random forest models using 200 decision trees (there were no significant improvements when using more trees), to maximize the depth of the decision trees (most of the leaves contain only one sample from the training data set), and to set the number of predictor variables considered for splitting the nodes at three. These parameters were chosen for all the models developed.

The prediction from a random forest model used for regression is the mean value of the predictions from all decision trees. For the direction of sea ice drift, each decision tree predicts a value between 0 and 360°. When averaging several predictions close to the northward direction, this can be an issue because values slightly higher than 0°and slightly lower than 360°can be averaged, possibly leading to a mean value close to the southward direction. In order to avoid this issue, the predictions from all decision trees (in degrees) were converted to complex numbers before averaging. Then, the average of complex numbers was converted into an angle in degrees. Furthermore, random forest models tend to predict less extreme values than the target variable because the mean value from all decision trees is used as the prediction. This should not be an issue for predicting the direction of sea ice drift due to the circular nature of directional data, but particularly low and high sea ice drift speed could be difficult to predict with random forest models.

The Canadian Arctic Archipelago is excluded from our study due to the different characteristics of sea-ice drift in this region (largely influenced by the presence of narrow channels and landfast ice) compared to the rest of the Arctic. Therefore, no data located in the Canadian Arctic Archipelago were used for training and evaluating the random forest models. Furthermore, the calibrated forecasts have been produced where the sea ice concentration predicted in the TOPAZ4 forecasts was larger than 10 %.

## 3.2 Training data sets

Only the TOPAZ4 forecasts starting on Thursdays are stored in the long-term archive, and the algorithms have therefore been trained using weekly data. The period from January 2018 to May 2020 was used for training the algorithms with SAR observations (the CMEMS SAR MOSAIC product has been available since January 2018). Due to the smaller number of available observations, a longer period (June 2013 to May 2020) was used for training the algorithms with buoy observations. While using a longer period increases the size of the training data sets, it can also introduce inconsistencies in the training data sets due to the constant development of the prediction systems (TOPAZ4 and ECMWF Integrated Forecasting System). Several training periods were tested between June 2012 and May 2020, and the chosen period from June 2013 to May 2020 seems to be optimal for predicting the direction of sea ice drift. However, using a shorter training period would have improved the forecasts for the speed of sea ice drift (figure S2 of the supplementary material). This is probably due to the smaller bias of TOPAZ4 sea ice drift speed in the recent years, which results from the negative trend of the sea-ice drift speed in TOPAZ4 (in contrast with IABP observations which show an acceleration, see figure S6 of the supplementary material). Nevertheless, we decided to use the same training period for the random forest models predicting the direction and speed of sea ice drift for consistency.

For the algorithms trained with buoy observations, the data from all the grid points where buoy observations were available within the TOPAZ4 domain have been used in order to create a large database, which results in about $1.5 \times 10^4$ data points for each model during the training period. However, a different approach was used for the algorithms trained with SAR observations. Taking into account all the grid points where SAR observations were available in the TOPAZ4 domain would result in using many highly correlated points for training the algorithms, which increases the probability of overfitting. In order to minimize this issue, but also taking into account a sufficient number of grid points, sensitivity tests were performed. For each

forecast of the training period (January 2018 - May 2020), a random selection without replacement of the grid points has been performed, and the selected grid points were added to the training data set (similar approaches were used by Gagne et al. (2014) and Loken et al. (2019) for calibrating precipitation forecasts). Random selections between 0.1 % and 100 % of the available grid points were tested, and the differences in mean absolute errors between the algorithms trained using all the available data and the algorithms trained using only a fraction of the available data were evaluated (figure S1 of the supplementary material). Due to the good performances of the models trained using 2 % of the available grid points, we decided to keep 2 % of the available grid points for training the models with SAR observations. This results in using data from about 5.5 x $10^4$ data points on average (between 5.2 x $10^4$ and 5.7 x $10^4$ data points depending on lead time). Moreover, decreasing the size of the training data sets reduces the computational cost of the algorithms. Furthermore, it is worth noting that the Arctic is not uniformly covered by SAR and buoy observations (figure 1), and that different regions have therefore different weights in the development of the algorithms.

## 3.3 Pre-processing of the data

In order to avoid overfitting, it is better to use predictor variables that are not highly correlated. This is why the speed and direction of sea-ice drift, as well as the wind speed and direction, have been used as predictor variables instead of the eastward and northward components. The speed of sea ice drift was determined from the great-circle distance calculated using the Haversine formula (equation 1) between the start and end locations during 24 hours. The initial great-circle course angle was used for the direction of sea ice drift, and was calculated using equation 2:

$$D = 2R \arcsin \left( \sqrt{\sin^2 \left( \frac{\varphi_{end} - \varphi_{start}}{2} \right) + \cos(\varphi_{start}) \cos(\varphi_{end}) \sin^2 \left( \frac{\lambda_{end} - \lambda_{start}}{2} \right)} \right) \tag{1}$$

$$\theta = \arctan2 \Big( \sin(\lambda_{end} - \lambda_{start}).\cos(\varphi_{end}), \ \cos(\varphi_{start}).\sin(\varphi_{end}) - \sin(\varphi_{start}).\cos(\varphi_{end}).\cos(\lambda_{end} - \lambda_{start}) \Big) \left( \frac{180}{\pi} \right) \tag{2}$$

where $arctan2$ represents the 4-quadrant inverse tangent function, $R$ is the Earth's radius, $\varphi$ and $\lambda$ represent the latitude and the longitude, and the subscripts "start" and "end" indicate the start and end locations. Furthermore, the wind speed and direction from ECMWF forecasts were calculated using equations 3 and 4, respectively, from the mean daily $u$ and $v$ components ($\overline{u}$ and $\overline{v}$ in equations 3 and 4). The computed wind direction is not the classic meteorological wind direction (direction from which the wind is blowing), but the opposite (direction the wind is blowing to) in order to be consistent with the direction of sea ice drift calculated using equation 2.

$$WS = \sqrt{\overline{u}^2 + \overline{v}^2} \tag{3}$$

$$WD = \arctan2(\overline{u}, \overline{v}) \left( \frac{180}{\pi} \right) \tag{4}$$

We also tested random forest models predicting the sea ice drift along the x and y axes of the TOPAZ4 grid using a different set of predictor variables (figure S12 of the supplementary material). For these models, the northward and eastward components of the ECMWF wind forecasts were used as predictors instead of the wind speed and direction, as well as the sea ice drift along the x and y axes from TOPAZ4 forecasts (which are provided by TOPAZ4 outputs) instead of the sea ice drift speed and direction. The direction and speed of sea ice drift were then calculated using the start and end locations of the sea ice for comparing those models with the ones directly predicting the direction and speed of sea ice drift. Relatively similar performances were achieved by these models for predicting the direction of sea ice drift, but these models had significantly worse performances for predicting the speed of sea ice drift (larger mean absolute errors of about 12.2 % and 13.7 % on average for the models trained with buoy and SAR observations, respectively).

## 3.4 Evaluation of the sea ice drift forecasts

While comparing the speed of sea ice drift in two data sets is straightforward, caution is needed when comparing the direction of sea ice drift in different data sets due to the circular nature of directional data. The direction errors were calculated using equation 5, where $D_x$ and $D_y$ are the two directions compared in degrees (between 0° and 360°). Furthermore, the correlation between different data sets has been assessed using the Pearson correlation coefficient for the speed, and using the circular correlation coefficient (equation 6) introduced by Fisher and Lee (1983) for the direction. In equation 6, $\bar{x}$ and $\bar{y}$ are the means of the variables x and y respectively. Similarly to the Pearson correlation coefficient, the value of the circular correlation coefficient varies between -1 and 1 (a null value indicating no correlation, 1 meaning a perfect correlation, and -1 showing a perfect anti-correlation).

$$\Delta D = D_x - D_y \Rightarrow Error = \begin{cases} \Delta D - 360, & \text{if } \Delta D > 180 \\ \Delta D + 360, & \text{if } \Delta D < -180 \\ \Delta D, & \text{otherwise} \end{cases} \tag{5}$$

$$R_c = \frac{\sum\limits_{i=1}^{n} \sin\left(x_i - \bar{x}\right) \sin\left(y_i - \bar{y}\right)}{\sqrt{\sum\limits_{i=1}^{n} \sin^2\left(x_i - \bar{x}\right) \sum\limits_{i=1}^{n} \sin^2\left(y_i - \bar{y}\right)}} \tag{6}$$

In this study, we used the Wilcoxon signed-rank test to assess the statistical significance of the differences between the absolute errors due to its suitability for non-parametric data (the absolute errors are not normally distributed) and paired observations (the same data set was used for evaluating the different models). We performed this analysis using the two-tailed hypothesis test and the significance level of 0.05.

## 3.5 Evaluation of the importance of predictor variables

In this study, the importance of the predictor variables was estimated using two different methods. First, the impurity-based feature importance was assessed. This method is based on the measure of impurity decreases (the mean squared error here) at all nodes in the random forest algorithm (the variables that often split nodes with large impurity decreases are considered important). It provides an assessment of the relative importance of the predictor variables, but is known for underestimating the importance of non-continuous predictors (Strobl et al., 2007).

In addition, the random forest models using all predictor variables have been compared to models in which one of the predictors was removed. This experiment was performed using the same method as the one used for determining the parameters of the random forest algorithms, which consists of training the models with data from about 80 % of the forecast start dates from the training periods, and using the remaining data for evaluating the forecasts (see section 3.1). By comparing the mean absolute errors of the different models, this method allows to determine if the predictor variables tend to improve or deteriorate the forecasts. Furthermore, it is worth noting that this method tends to underestimate the importance of highly correlated predictors since similar information is provided to the algorithm when one of the correlated predictors is removed.

## 4 Results

### 4.1 Evaluation of daily SAR observations

The CMEMS SAR MOSAIC product has been compared to buoy observations during the period from January 2018 to December 2020 in figure 2. The Pearson correlation coefficient is 0.80 for the speed, and the circular correlation coefficient is 0.84 for the direction. The SAR observations have relatively low biases (the mean error is 2.8 degrees for the direction and -0.14 km / day for the speed). Furthermore, the mean absolute error is 22.0 degrees for the direction and 2.1 km / day for the speed (the mean speeds are 7.46 and 7.60 km / day for the SAR and buoy observations, respectively). The root mean square error is 36.8 degrees for the drift direction and 3.6 km / day for the drift speed. While these errors are considerable, the large number of SAR observations compared to buoy observations makes this product potentially suitable for machine learning applications. However, we consider these errors too large for evaluating the performances of the sea ice drift forecasts using only these observations. Therefore, the performances of sea ice drift forecasts have been evaluated using buoy observations, and the SAR observations have been used to study the spatial variability of the forecast performances.

### 4.2 Evaluation of the calibrated forecasts

The performances of the calibrated forecasts have been evaluated and compared to those from the TOPAZ4 prediction system during the period from June 2020 to May 2021 using buoy observations (figures 3). For predicting the direction of sea ice drift, the models trained with buoy observations significantly outperform the TOPAZ4 prediction system and the models trained with SAR observations for all lead times, except 10 days. On average, the calibrated forecasts produced by these models have a mean absolute error about 8.0 % lower than TOPAZ4 forecasts. The models trained with SAR observations significantly

outperform the TOPAZ4 prediction system for lead times up to 5 days, and reduce the mean absolute errors by 3.3 % compared to TOPAZ4 forecasts. However, the TOPAZ4 prediction system slightly outperform the models trained with SAR observations for lead times from 8 to 10 days, though the differences are not statistically significant. Moreover, the fraction of forecasts improved by the calibration is, on average, larger for the models trained with buoy observations (55.7 %) than for the models trained with SAR observations (52.9 %). Furthermore, the correlation between the forecasts and the buoy observations is improved by both calibration methods for lead times up to 7 days, and deteriorated for longer lead times.

For the speed of sea ice drift, the models trained with buoy observations have the best performances for all lead times. They significantly outperform the TOPAZ4 system and the models trained with SAR observations for all lead times, except 4 days for which the difference with the TOPAZ4 system is not statistically significant. The forecasts from the models trained with SAR observations have slightly larger mean absolute errors than TOPAZ4 forecasts for lead times up to 5 days, but significantly outperform TOPAZ4 forecasts for longer lead times. On average, the mean absolute error is reduced by 7.1 % and 2.5 % by the calibration for the models trained with buoy and SAR observations, respectively. The fraction of forecasts improved is, on average, slightly larger for the models trained with buoy observations (53.4 %) than for the models trained with SAR observations (53.1 %). Moreover, the correlation between the buoy observations and the forecasts is improved by both calibration methods.

The spatial variability of the fraction of forecasts improved by the calibration has been analyzed using SAR observations as reference in order to use as many observations as possible (figures 4, 5, 6, 7), though the grid points with less than 20 SAR observations during the period from June 2020 to May 2021 have been excluded from this analysis. The number of SAR observations per grid cell used for this comparison has been mapped in figure 1 d). Overall, both calibration methods perform relatively well for predicting the direction of sea ice drift in the Central Arctic for lead times up to 5 days (figures 4 and 5). However, the fraction of forecasts improved decreases with increasing lead times, and both calibration methods have relatively poor performances in the Beaufort, Chukchi, and East Siberian seas. Furthermore, the models trained with buoy observations perform better than the models trained with SAR observations in most of the area taken into account in this analysis.

For the speed of sea ice drift, the models trained with SAR observations perform better than the models trained with buoy observations in most of the area analyzed. The models trained with buoy observations have particularly poor performances compared to TOPAZ4 near the Greenland and Russian coastlines (figure 6), while the models trained with SAR observations perform better in these areas (figure 7). It is worth noting that most of the buoys taken into account for evaluating the forecasts in figure 3 are not located in the areas where the models trained with buoy observations have poor performances, which likely explains the better performances of the models trained with buoy observations compared to the models trained with SAR observations in figure 3.

## 4.3 Importance of predictor variables

For both calibration methods, the most important variable for predicting the drift direction is the sea ice drift direction from TOPAZ4 forecasts, followed by the wind direction from ECMWF forecasts (figure 8). On average, the relative importance of sea ice drift direction forecasts is about 1.4 and 1.5 times larger than the one from wind direction forecasts for the models

trained with buoy and SAR observations, respectively. The sum of the relative importances of these two variables represent, on average, about 46 and 41 % of the sum of all relative importances for the models trained with buoy and SAR observations, respectively. However, the relative importances of these two variables decrease with increasing lead times.

Similarly, the sea ice drift speed from TOPAZ4 is the most important variable for predicting the speed of sea ice drift, followed by the wind speed from ECMWF forecasts. On average, the relative importance of sea-ice drift speed forecasts is about 1.7 and 2.2 larger than the one from wind speed forecasts for the models trained with buoy and SAR observations, respectively (figure 8). For the models predicting the speed of sea ice drift, the sum of the relative importances of these two variables represent, on average, about 40 % of the sum of all relative importances for both calibration methods. Furthermore, the relative importances of these two variables also decrease with increasing lead times.

On average, the mean absolute errors are reduced by all predictors for the direction and speed of sea ice drift in both calibration methods (figure 9), though some predictor variables do not improve the forecast accuracy for all lead times. While the sea ice concentration from TOPAZ4 forecasts and from the observations during the initialization of the forecasts are correlated, removing one of these variables decreases the accuracy of most random forest models. Therefore, we decided to keep both variables, even if the importances of these variables are probably underestimated due to this correlation. Furthermore, we also tested using the day of year as an additional predictor variable (figure S7 of the supplementary material), but adding this variable tends to deteriorate the forecast accuracy for most models, so we decided to discard this variable.

For the models predicting the direction of sea ice drift, removing the drift direction from TOPAZ4 forecasts increases the mean absolute error between 1.1 and 6.7 degrees depending on the lead time and the observations used for the target variable. This is much larger than the differences in mean absolute error when the wind direction from ECMWF forecasts is removed (between 0.1 and 2.2 degrees). For the models predicting the speed of sea ice drift, removing the drift speed from TOPAZ4 forecasts increases the mean absolute error between 0.041 and 0.444 km /day depending on the lead time and the observations used for the target variable. This is also much larger than the differences in mean absolute error when the wind speed from ECMWF forecasts is removed. Surprisingly, removing the wind speed forecasts slightly reduces the mean absolute error (difference of 0.005 km / day) for the model predicting the speed of sea ice drift for a lead time of 4 days trained with SAR observations. For the other models predicting the speed of sea ice drift, removing the wind speed forecasts increases the mean absolute error between 0.001 and 0.127 km / day. Furthermore, the mean absolute errors for the speed of sea ice drift are also considerably reduced by adding the sea ice thickness forecasts from TOPAZ4 (between 0.011 and 0.098 km / day), probably due to the anti-correlation between sea ice thickness and sea ice drift speed (Yu et al., 2020).

## 5 Discussion and conclusion

The characteristics and performances of the calibrated forecasts developed in this study depend on the observations used for training the algorithms, as well as the data sets used for the predictor variables. The 24-hour mean composites of drift vectors provided by the CMEMS SAR MOSAIC product have been used for training some algorithms due to the fixed time spans of this data set, the large number of available observations, and the relatively high spatio-temporal resolution compared to sea

ice drift products developed from passive microwave observations (Lavergne et al., 2010; Girard-Ardhuin and Ezraty, 2012; Tschudi et al., 2020). However, the spatial resolution (10 km) and the time spans (24 hours) of this product prevent to develop

high-resolution calibrated forecasts. Furthermore, the averaging process of drift vectors introduces significant uncertainties in the product (figure 2), which represent a limitation for developing accurate calibrated forecasts. Buoy observations are more accurate, but the relatively low number of available observations is a limitation for the development of random forest models. Both methods also have common limitations such as the heterogeneous spatio-temporal sampling of the buoy and SAR observations. The random forest algorithms are more influenced by areas often covered by sea ice drift observations

than by areas with a poor coverage in these observations. This could potentially explain some of the spatial variability of the performances of the calibrated forecasts. Overall, the calibrated forecasts have their best performances in the Central Arctic where most of the training data are located.

The forecasts used for the predictor variables are produced by operational models (ECMWF Integrated Forecasting System and the TOPAZ4 prediction system) which are constantly developed. The development of these systems could affect the

355 performances of the random forest algorithms due to changes between the different versions of the models. This issue is more important for the models trained with buoy observations due to the longer period used for training these models. Moreover, TOPAZ4 does not reproduce the recent acceleration of sea ice drift as already reported by Xie et al. (2017), and the bias of TOPAZ4 sea ice drift speed has changed during the studied period (figure S6 of the supplementary material). This probably affects the performances of the random forest models trained with buoy observations due to their relatively long training period,

and using a shorter training period would have improved the performances for predicting the speed of sea ice drift (figure S2 of the supplementary material). Furthermore, while only the TOPAZ4 forecasts starting on Thursdays were used for training the random forest algorithms (only these forecasts are stored in the long-term archive), the operational forecasts have been produced daily. Because data assimilation is only performed on Thursdays, this could be an issue when producing forecasts not starting on Thursdays (the weights of the different predictor variables might not be optimal).

Despite these limitations, it has been shown that the random forest models trained with buoy observations outperform TOPAZ4 forecasts for both the speed and direction of sea ice drift, except for 10-day lead time for the direction of sea ice drift. The models trained with SAR observations significantly outperform TOPAZ4 forecasts for lead times up to 5 days for the direction of sea ice drift, but have similar performances as TOPAZ4 forecasts for longer lead times. For the speed of sea ice drift, the models trained with SAR observations have slightly larger mean absolute errors than TOPAZ4 for lead times up

to 5 days, but significantly outperform TOPAZ4 forecasts for longer lead times. On average, the mean absolute errors for the direction of sea ice drift are 8.0 % and 3.3 % lower than in TOPAZ4 forecasts for the models trained with buoy and SAR observations, respectively. For the models predicting the speed of sea ice drift, the mean absolute errors are reduced by 7.1 % and 2.5 % for the models trained with buoy and SAR observations, respectively. The lower errors of the models trained with buoy observations, despite their smaller training data sets, show that the accuracy of the observations used for the target

variables plays a crucial role in the performances of the calibrated forecasts.

The spatial analysis of the forecast performances using SAR observations as reference has shown that the models trained with buoy observations outperform the models trained with SAR observations for predicting the direction of sea ice drift in

most of the area analyzed in this study (figures 4 and 5). However, for the algorithms predicting the speed of sea ice drift, the models trained with SAR observations have better performances in most of the area analyzed in this study (figures 6 and 7). The models trained with buoy observations have particularly poor performances for predicting the speed of sea ice drift near the Greenland and Russian coastlines. This is likely due to the low number of buoy observations available for training the algorithms in these areas and the particular characteristics of these areas (high drift speed along the Greenland East coast and presence of landfast ice along the Russian coast).

In order to reduce forecast errors from numerical prediction systems, calibration procedures can be applied, although their performances depend on the number and accuracy of the observations used as target variables. The increasing amount of satellite data available and the improvements in sea ice remote sensing, as well as the development of new statistical approaches, enhance the potential of calibration techniques for sea ice forecasting. This should contribute to improve the accuracy of sea ice forecasts delivered to maritime end-users.

*Code and data availability.* Buoy observations are available on the International Arctic Buoy Programme (IABP) website (https://iabp.apl.uw.edu). Synthetic-aperture radar observations and TOPAZ4 forecasts are available on the Copernicus Marine Environment Monitoring Service server (https://resources.marine.copernicus.eu/). The OSI-SAF sea ice concentration observations can be downloaded from the MET-Norway FTP server (ftp://osisaf.met.no/ reprocessed/ice/conc/v2p0) until 2015, but the data after 2015 are not publicly available. A licence is needed to download the wind forecasts from the European Centre for Medium-Range Weather Forecasts (ECMWF). Furthermore, the codes used for this analysis are available in the following Github directory: "https://github.com/cyrilpalerme/Calibration_of_sea_ice_drift_forecasts/".

*Author contributions.* C.P. conducted the analysis and wrote the majority of the manuscript. C.P. and M.M. designed the research and contributed to the discussions of the results.

*Competing interests.* The authors declare that they have no conflict of interest.

*Acknowledgements.* The authors would like to thank John Bjørnar Bremnes, Siri Sofie Eide, Roberto Saldo, and Thomas Lavergne for valuable discussions, as well as Atle Macdonald Sørensen for maintaining the production of sea ice concentration observations from AMSR2 and SSMIS at the Norwegian Meteorological Institute. The authors received funding from the Copernicus Marine Environmental and Monitoring Service Arctic Marine Forecasting Center and IcySea project through Mercator Océan, as well as, from the SALIENSEAS project funded by the Norwegian Research Council contract number 276223. This is a contribution to the Year of Polar Prediction (YOPP), a flagship activity of the Polar Prediction Project (PPP), initiated by the World Weather Research Programme (WWRP) of the World Meteorological Organisation (WMO). Finally, we thank the three reviewers for their comments which significantly improved the manuscript.

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

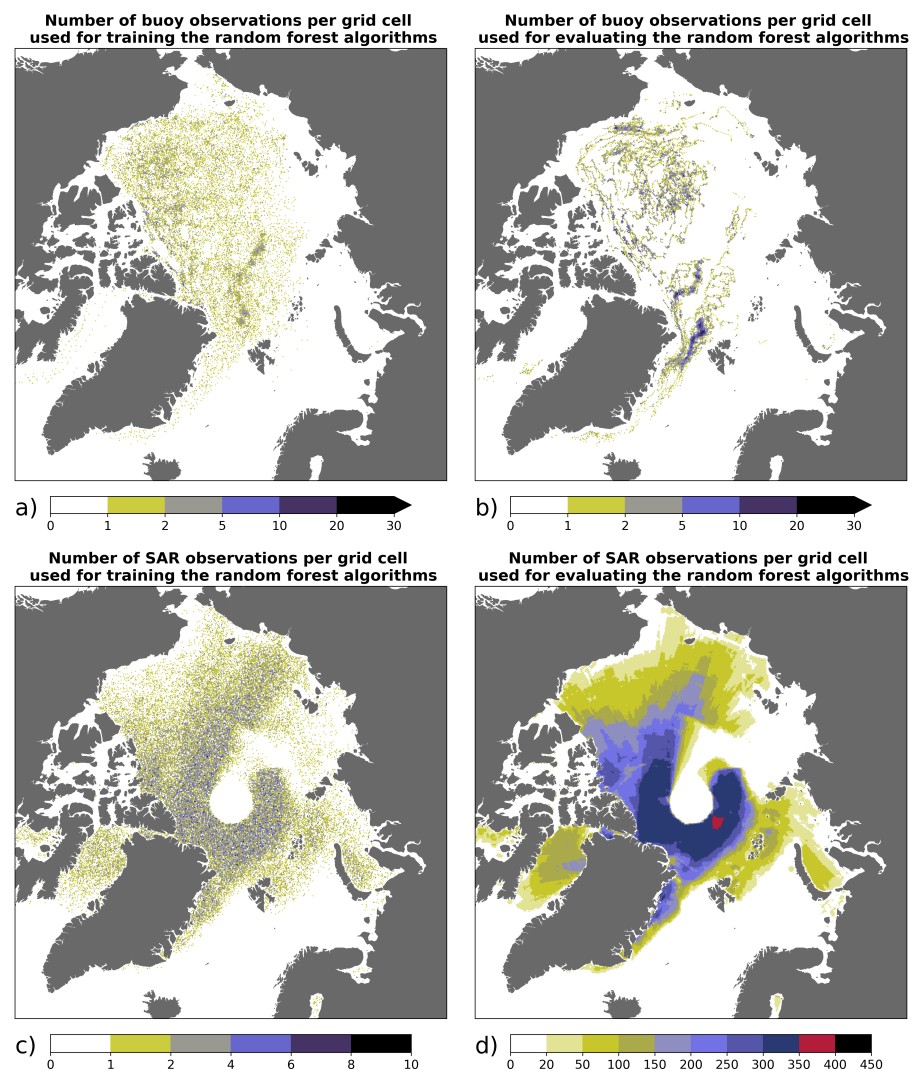

**Figure 1.** a) Number of buoy observations per grid cell used for training the random forest algorithms during the period from June 2013 to May 2020. b) Number of buoy observations per grid cell used for evaluating the random forest algorithms during the period from June 2020 to May 2021. c) Number of SAR observations per grid cell used for training the random forest algorithms during the period from January 2018 to May 2020. d) Number of SAR observations per grid cell used for evaluating the random forest algorithms during the period from June 2020 to May 2021. These four maps show the number of observations for a lead time of one day (similar results have been found for other lead times).

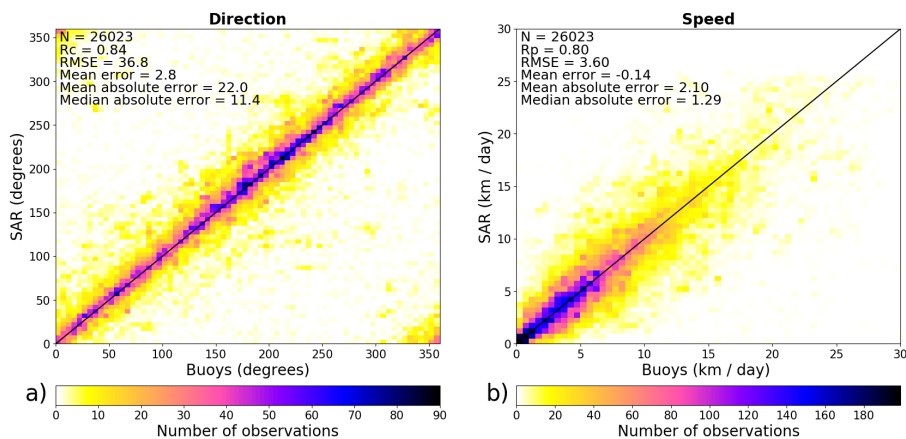

**Figure 2.** Evaluation of the CMEMS SEAICE_GLO_SEAICE_L4_NRT_OBSERVATIONS_011_006 product (MOSAIC version 2.0) for the sea ice drift direction (a) and speed (b) during the period from January 2018 to December 2020. The axes are bounded to 30 km / day for the speed for clarity, but the speed of 73 buoys exceed 30 km / day. However, these buoys were taken into account when calculating the statistics shown on the figures (N: number of samples, Rc: circular correlation coefficient, Rp: Pearson correlation coefficient, RSME: root mean square error). The color scales represent the number of observations in each bin of 5 degrees for the direction and 0.5 km / day for the speed.

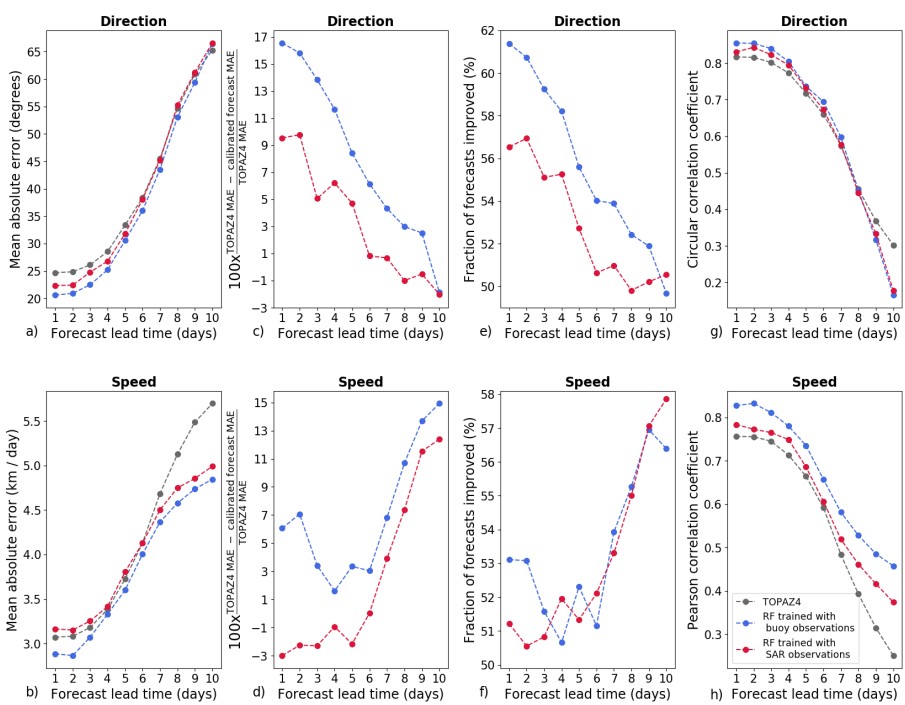

**Figure 3.** Evaluation of the performances of the calibrated forecasts produced using random forest (RF) algorithms and the TOPAZ4 forecasts during the period from June 2020 to May 2021. IABP buoy observations have been used as reference. Mean absolute errors (MAE) of the forecasts for the direction (a) and the speed (b), and relative improvement ($100 \times \frac{\text{TOPAZ4 MAE - calibrated forecast MAE}}{\text{TOPAZ4 MAE}}$) for the direction (c) and the speed (d). Fraction of calibrated forecasts with lower absolute errors than the TOPAZ4 forecasts for the direction (e) and the speed (f). Circular and Pearson correlation coefficients between the forecasts and the buoy observations for the direction (g) and the speed (f), respectively.

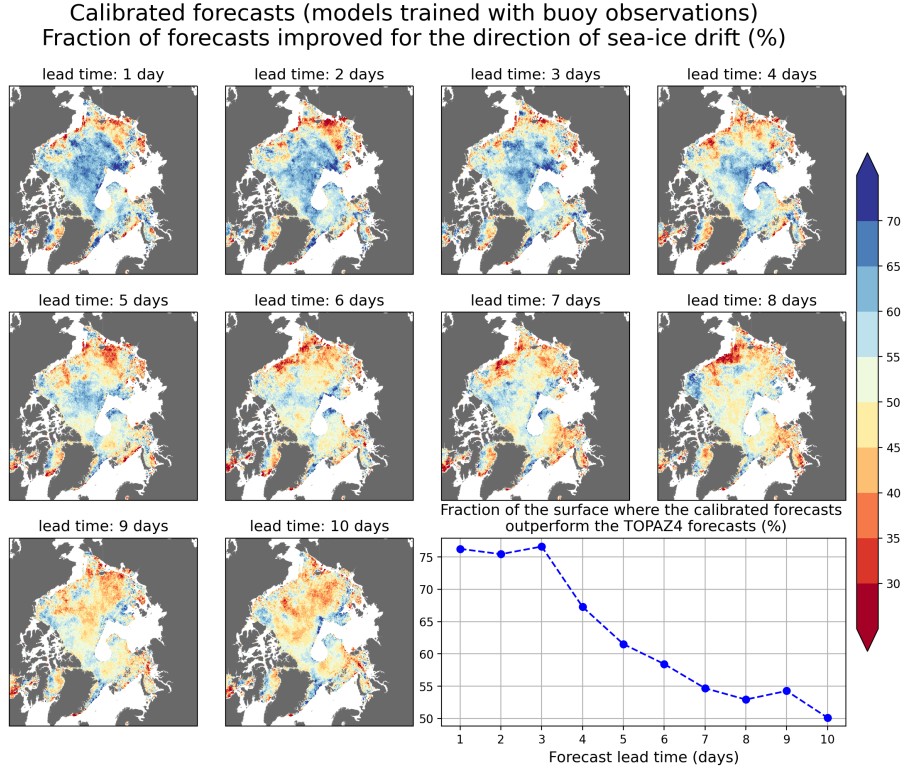

**Figure 4.** Fraction of calibrated forecasts produced by the models trained with buoy observations which outperform the TOPAZ4 forecasts for the direction of sea ice drift during the period from June 2020 to May 2021. Daily SAR observations have been used as reference. The graph in the lower right corner shows the fraction of the surface where the fraction of calibrated forecasts outperforming the TOPAZ4 forecasts is higher or equal to 50 %.

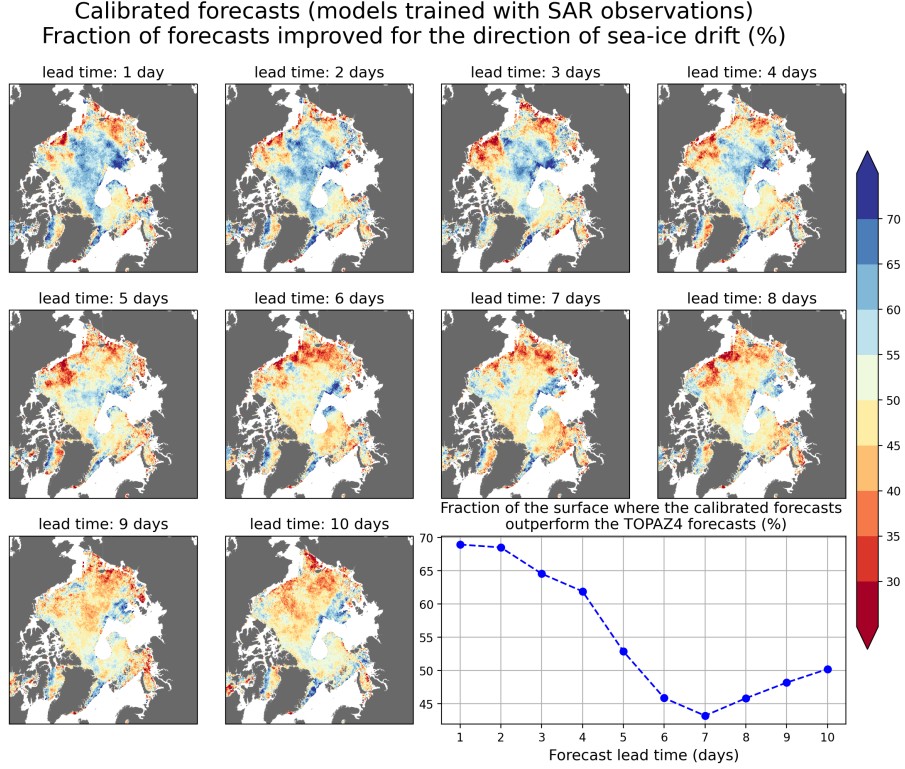

**Calibrated forecasts (models trained with SAR observations)**
**Fraction of forecasts improved for the direction of sea-ice drift (%)**

**Figure 5.** Fraction of calibrated forecasts produced by the models trained with SAR observations which outperform the TOPAZ4 forecasts for the direction of sea ice drift during the period from June 2020 to May 2021. Daily SAR observations have been used as reference. The graph in the lower right corner shows the fraction of the surface where the fraction of calibrated forecasts outperforming the TOPAZ4 forecasts is higher or equal to 50 %.

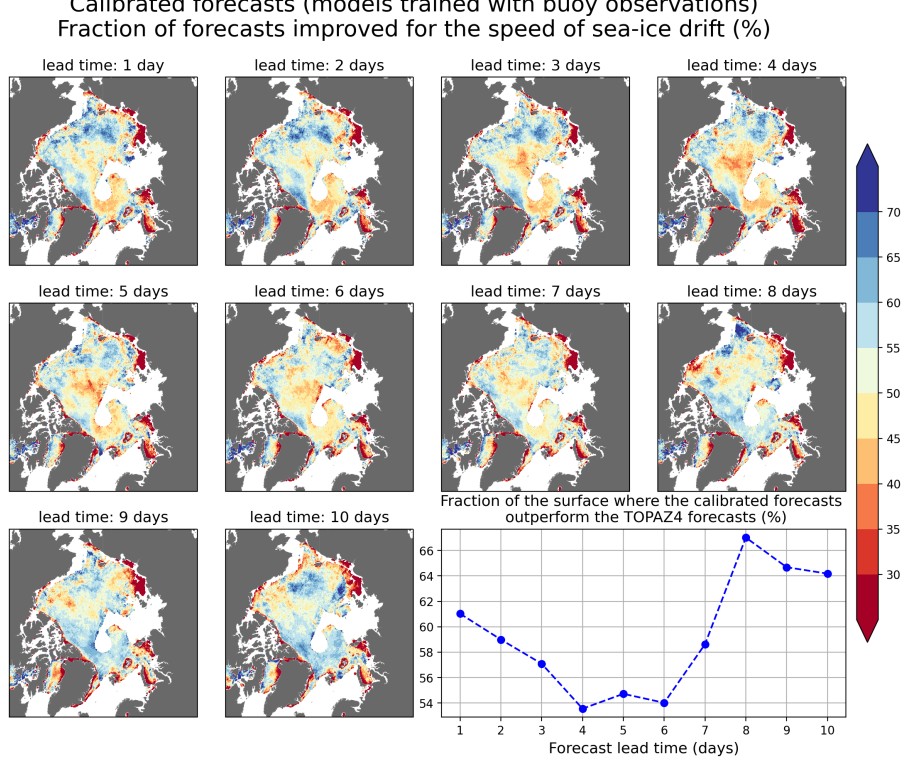

**Figure 6.** Fraction of calibrated forecasts produced by the models trained with buoy observations which outperform the TOPAZ4 forecasts for the speed of sea ice drift during the period from June 2020 to May 2021. Daily SAR observations have been used as reference. The graph in the lower right corner shows the fraction of the surface where the fraction of calibrated forecasts outperforming the TOPAZ4 forecasts is higher or equal to 50 %.

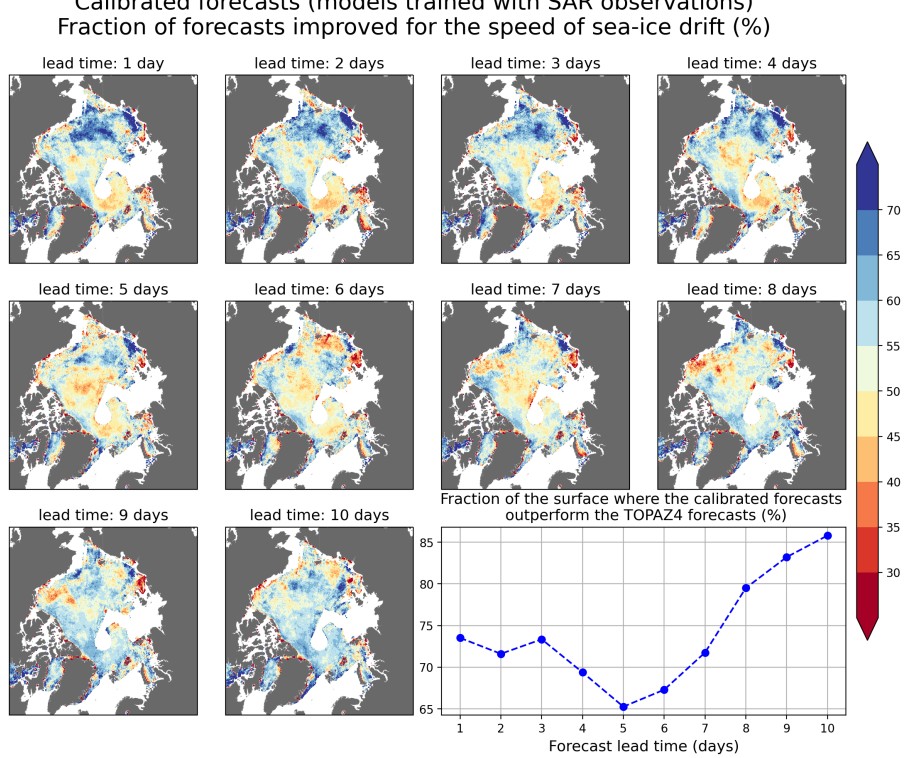

**Figure 7.** Fraction of calibrated forecasts produced by the models trained with SAR observations which outperform the TOPAZ4 forecasts for the speed of sea ice drift during the period from June 2020 to May 2021. Daily SAR observations have been used as reference. The graph in the lower right corner shows the fraction of the surface where the fraction of calibrated forecasts outperforming the TOPAZ4 forecasts is higher or equal to 50 %.

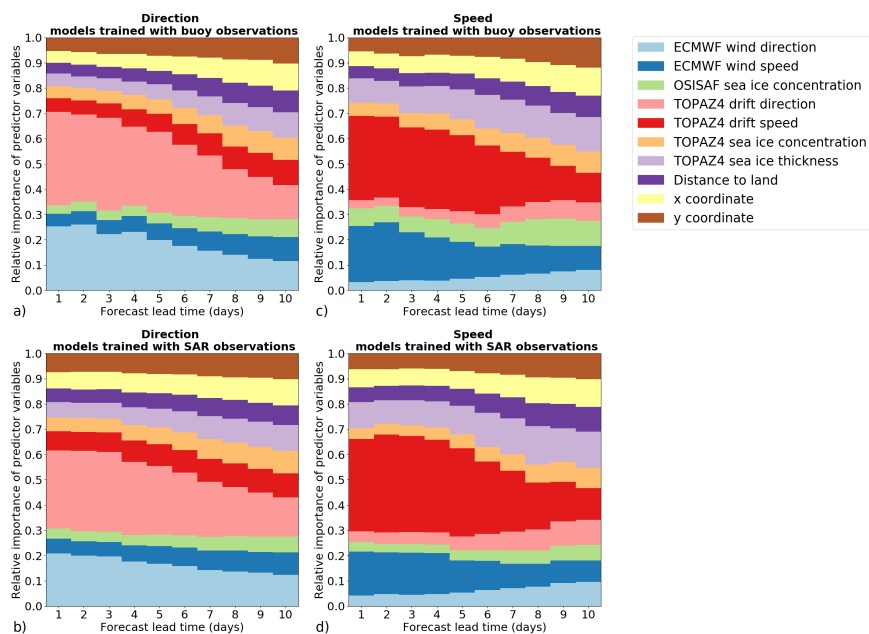

**Figure 8.** Relative importance of the predictor variables for the direction (a, b) and the speed (c, d) of sea ice drift assessed using the impurity-based feature importance method.

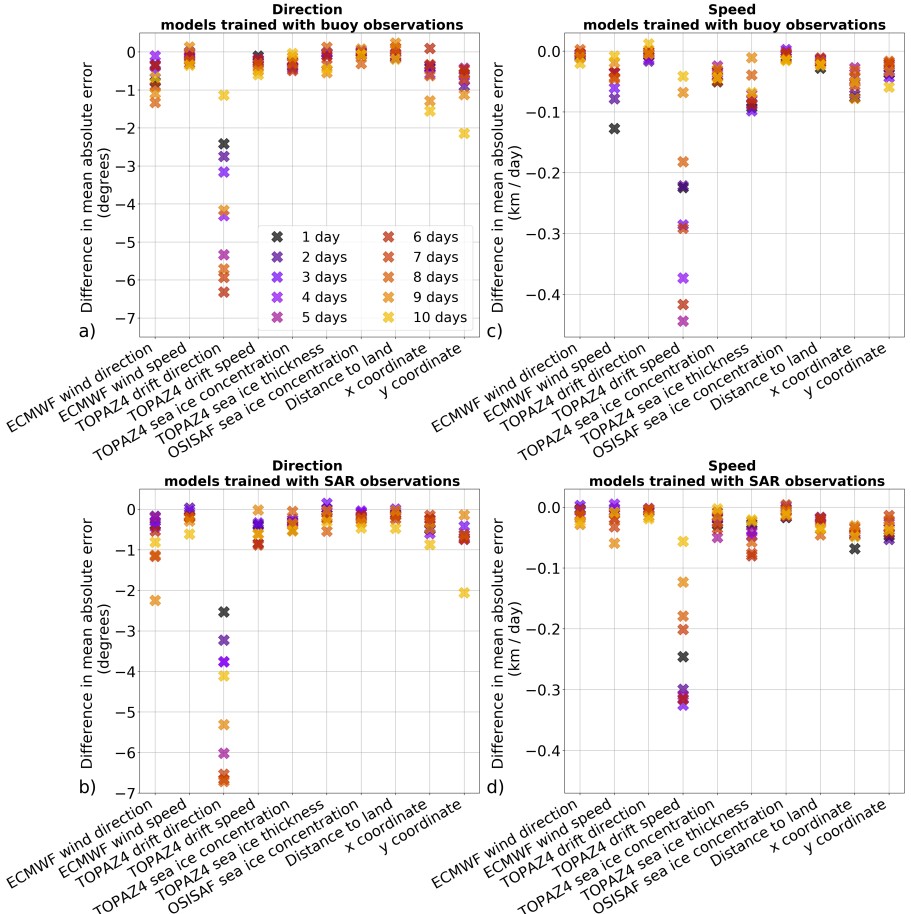

**Figure 9.** Differences in mean absolute error when one of the predictor variables is not used in the random forest models for the direction (a, b) and speed (c, d) of sea ice drift. The results are shown for the models trained with buoy observations (a, c), and for the models trained with SAR observations (b, d). The lead times are indicated in the legend of figure a). The differences represent the subtraction between the performances of the models using all the predictor variables and the models in which one predictor variable was not used. Therefore a negative value means that adding the variable in the algorithm improves the forecasts.