# Peer review of "Calibration of sea ice drift forecasts using random forest algorithms"

_The Cryosphere, 2021_

## Referee Comment (RC3)

Title: Calibration of sea ice drift forecasts using random forest algorithms
Authors: Cyril Palerme and Malte Müller

This papers present short-term (1-10 days) forecasts of sea ice drifts using a random forest (AI) algorithm and a comparison of the AI forecasts with those of the operational ice-ocean prediction system TOPAZ. The models were trained using buoy or radarsat-derived sea-ice drifts. Predictors include short-term forecast ice speed and angle, wind speed and angle and ice thickness. Results show that the AI forecasts are more skillful than those of TOPAZ irrespective of the training data set. Furthermore, the model trained using sea ice buoys is more skillful in predicting sea ice drift for all lead-time when compared with the model trained with radarsat ice drifts.

The paper addresses an interesting question. The use of AI in sea-ice forecasting is relatively new and for this reason, this is a welcome contribution. The paper however is not well written, the introduction is succinct and does not place the work in the context of previous effectively, the model section is entirely missing and there is relatively little discussion of the pre-processing of the input data and its impact on the forecast skill (a factor that is at least equally important as the AI algorithm in producing a skillful model).

I recommend that the paper be accepted for publication after the comments below have been addressed (i.e. not rebutted).

**Major Points**:

1- The paper must be substantially edited/restructured.
   i)     The introduction is vague, there is a lot of name-dropping but it does not present an in-depth description of the previous work that is required to fully appreciate the content of the paper. I suggest that the authors review the literature more in-depth and revise the introduction substantially, or add a third co-author that works more closely in the field of sea ice forecasting.
   ii)    The use of the present perfect to describe the data is odd. "Satellite sea-ice drift observations … have been used…" (Line 47, 56, etc.). It sounds as though the authors are speaking of previous work by other authors when they are speaking of their own work being presented. The use of the present tense is much more engaging for the reader, or at least the simple past. I.e. We use (used) satellite sea-ice drifts observations…" . These are two examples; there are many more in the paper.
   iii)   Line 67: "Section 2.2: Data used for the predictor variable". The model used as a reference for the evaluation of the AI models (i.e. TOPAZ) is included here, yet it is not a predictor variable. The forecasted sea ice thickness, concentration, drift speed and angle are all predictor variables but are not described in this section. Only the 10-m wind speed is discussed.

iv)     TOPAZ is described only very succinctly. It does not say which sea ice model is used, whether there is an ice thickness distribution included, the grid on which the equation are solved, etc.

2- The model is section is entirely missing. A mathematical description of the random forest model must be given because AI is relatively new in the field of short-term sea ice forecasting and more simply for the sake of completeness. The reader should not have to read other papers about random forest in order to fully appreciate the content of the current work.

3- Some pre-processing was done to the data. E.g. the authors used speed and angle rather than latitudinal and meridional components; two different models for speed and angle were proposed. All these decisions leads to improvements in the forecast. Was there any more pre-processing done to the data to improve skill? What was the improvement in the forecast skill using these pre-processing techniques? A few sentences should be included in the discussion about this in section 4.3. I would call this section "Pre-processing of the data".

4- Line 215: A model trained within the Arctic Ocean proper should not be used to predict sea-ice drift in the land-lock sea ice of the Canadian Arctic Archipelago. This is an entirely different dynamical regime. This results and associated discussion should be removed from the paper and from the abstract. Or at least not given such an important presence.

**Minor Points**:

Line 13-14: Sea ice conditions in the Arctic do not change increasingly faster because of increase in ice drift speed. Increase sea ice drift speed is one such change associated with arctic climate change, but it is not the cause. The cause is thinning of sea ice associated with warmer air temperature, change in cloud phase and its impact on the radiative fluxes at the surface, increased ocean heat flux that interacts closely with sea ice on the shallow arctic shelves, increased storminess in the Arctic, etc

Line 61: Why only use sea ice drift speed lower than 5km per day? The mean speed in the Arctic Ocean is 5km /day or ~5cm/sec. It seems that a large amount of data is being ignored without acknowledging it or without providing a rationale for doing so.

Line 63: "…have been projected onto the grid used in the TOPAZ4 system". This is not useful information. What grid is used in TOPAZ4? Tri-polar? Curvi-linear? Cube-sphere? I see now that this has been defined later in the paper on Line 103. The grid must be defined when it is first discussed. Is it a Cartesian grid? Or Lat/Lon?

Line 79: Which ocean observations are assimilated?

Line 86: When did the switch to higher resolution happened?

Line 95: No new paragraph here. "… where R is the Earth's radius, lamda and phi are the…"

Equ 4: Unusual notation. arctan(v/u)?

Line 121: Should it be "data points" instead of "data sets"?

Line 165, Equ. 5: Why Case #3 in Equ. 5? Don't Case #1 and #2 above cover all cases?

Line 169-171: This is "Method" material that was already covered earlier. It should be moved to the method section.

Line 191. "Moreover the fraction of forecasts improved by the calibration is, on average, larger for the models trained with buoy observations (57.0 %) than for the models trained with SAR observations (54.8 %)". Is this really statistically significant? Errors are provided throughout the paper but it does not transpire in the discussion. The errors should used to assess whether the improvements are significant or not.

Line 197: "The fraction of forecast improved is, on average, slightly larger for the models trained with SAR observations (55.3 %) than for the models trained with buoy observations (54.9 %)."Again, is this statistically significant?

Line 222: The fraction of data used in the training and validation of the model belongs to the Method section.

Line 225-230: Repetitive. This was already mentioned in the Method section.

Line 236: Sea ice thickness does not change very much in 10 days. I suspect the ice thickness at t=0 would be equally skillful. This should be mentioned.

Section 4.3: The discussion does not present a quantitative assessment of the predictive skill of each predictor. A more quantitative discussion should be provided.

Figure 1: Colorbar for the d panel should be changed to avoid saturation.

Figure 4: Units for sea ice drift should be km/day or ideally cm/sec. It should not be m/day.

Bruno Tremblay
McGill University

---

## Author Comment (AC1)

We would like to thank the reviewers for their comments which helped us to improve the quality of the manuscript. Please find below our responses to the reviewer's comments.

**Reviewer 1**
* * *
**# Summary**

Palerme and Müller use random forest regression to predict Arctic sea-ice drift speed and direction from a set of predictors that contains besides dynamical sea-ice drift forecasts (TOPAZ4) also wind forecasts, geographical coordinates, sea-ice concentration and thickness, and distance from land. Using both buoy and satellite-derived drift for training and evaluation, the authors find that the predicted drift slightly outperforms the original TOPAZ4 drift forecasts at all lead times considered (1-10 days); mean absolute errors are reduced by roughly 5-10%. In my view the study is very relevant and innovative, scientifically sound, and well presented. What I think deserves additional effort is to illuminate more clearly what happens within the "black box" of the random forecast algorithm, for example, which of the predictands are picked how often to split nodes, what the output resolution of the individual trees is, how the predictands "modify" the TOPAZ4 drift forecasts, how that compares to simpler bias corrections, and how such characteristics change with lead time. With more explanations along these lines, the article could help readers (including myself) to better understand how the approach really functions, thereby providing an educational example how ML methods can help us to enhance predictions beyond the direct outputs of numerical models. In summary, I recommend publication of this work in The Cryosphere subject to minor(-to-major) revisions as detailed in the following.
* * *
**#** Specific comments**

Regarding the term "calibration": In my view it would be helpful to clarify in how far the presented approach is a "calibration" of dynamical model-based drift forecasts. Typically, calibration in this context means to use raw dynamical model forecasts and to modify them in some systematic way, e.g., to remove model biases. However, here the TOPAZ4 drift forecasts are used qualitatively in the same way as the other predictands, which appears to be a conceptual deviation from the standard calibration approach and leads to interesting questions. For example, would there be ways to formulate the random forecast algorithms such that they are explicitly used to modify the raw TOPAZ4 drift forecasts rather than predicting the drift "from scratch"? Or is that basically equivalent to the way it's currently being done, treating the TOPAZ4 drift just like any other predictand? It would be good to provide some clarification and/or discussion in this regard.

All the predictors are similarly provided to the random forest algorithms (it is not possible to explicitly define some predictors as more important than others before training the models). However, the most relevant predictors will be used more often to split the nodes, and will have a more important role in the predictions than other predictors. Furthermore, we agree that the meaning of the term "calibration" here differs from systematic bias corrections. Nevertheless, it is commonly used to describe weather forecasts produced using machine learning techniques, including random forests based on similar approaches as our study (for example: Gagne et al., 2014; Loken et al., 2019; Hill et al., 2020). Therefore, we have decided to keep the term "calibration" in the manuscript.

P2L47+56: "... have been used for training some random forest algorithms ...": First, from these sentences it is at first not clear that you are not talking about previous work, but that this is what has been done in the present study. Second, the "some" sounds very vague, maybe you can refer here to Sect. 3.2.

We agree that it was not clear in the text, and we have replaced these sentences by:

"In this study, satellite sea ice drift observations from the CMEMS product named SEAICE\_GLO\_SEAICE\_L4\_NRT\_OBSERVATIONS\_011\_00675 (MOSAIC version 2.0, hereafter referred as CMEMS SAR MOSAIC product) were used for training some random forest algorithms (see section 3), as well as for analyzing the spatial variability of the performances of sea ice drift forecasts."

and

"In addition, data from the International Arctic Buoy Programme (IABP) were also used for training some random forest algorithms (see section 3), as well as for evaluating the SAR observations and the sea ice drift forecasts."

**Sect. 2.2.: I think it would help to make very clear here that the TOPAZ4 drift forecasts are the basic ingredient here, but that other predictands are added and actually treated in the same way as the TOPAZ4 drift forecasts within the random forest algorithms, see my previous remarks.**

A more detailed description of the random forest method has been added in section 3.1 of the revised version of the paper which describes how the predictor variables are selected to split the nodes:

"Random forest algorithms consist of an ensemble of decision trees used for regression or classification tasks (Breiman, 2001). In order to avoid overfitting (meaning that the models learn from noise in the training data), independent decision trees must be developed. The independence of decision trees is ensured by using different subsets of the training data set for developing each decision tree, as well as by randomly selecting a fraction of the predictor variables at each node (the node is then split using the variable maximizing a dissimilarity metric among the selected predictors). Each decision tree is trained with a data set created using the bootstrap method, which consists of randomly selecting samples from the original training data with replacement for creating a new data set of the same size as the original one. This results in using about 63 % of the samples from the original data set for training each decision tree.

In this study, random forest models were developed for regression using the Python library Scikit-learn-0.23.2 (Pedregosaet al., 2011), and the mean squared error was used to measure the quality of the splits. Different models were developed for predicting the direction and speed of sea ice drift, as well as for different lead times (1 to 10 days). Moreover, two sets of models were developed using target variables either from buoy displacements or from SAR observations. Therefore, 20 different models were developed using buoy displacements, and 20 other models were developed using SAR observations. "

P3L79-80: "while TOPAZ4 forecasts are produced daily, only the forecasts starting on Thursdays are initialized using data assimilation": This sounds as if the forecasts starting on other days than Thursdays would not at all be affected by data assimilation, but I assume that they are affected by previous data assimilation, that is, from the last Thursday (and earlier), right? So I would say they are still "initialized", just not with particularly timely observations.

It is right that the TOPAZ4 forecasts are affected by the previous data assimilation (last Thursday). The sentence:

"However, while TOPAZ4 forecasts are produced daily, only the forecasts starting on Thursdays are initialized using data assimilation and stored in the long-term archive."

has been replaced by:

"While TOPAZ4 forecasts are produced daily, data assimilation is only performed on Thursdays, and only the forecasts starting on Thursdays are stored in the long-term archive."

**P4L91:** "The initial bearing on the great-circle path": From the context one can guess what is meant by "bearing" here, but is this word really correct?**

Thanks for this comment. We have checked this, and *"initial great-circle course angle"* seems to be the most common term. We have used this term in the revised version of the paper.

**P5L120: "as independent data sets": Please clarify what you mean here exactly by "independent".**

We meant that we used all the grid points with buoy observations similarly for training the random forest algorithms. However, some of the grid points could be spatially correlated, and the term "independent" would not be appropriate. We have decided to remove the term "independent" here.

**P5L121-133: Given that, if I understand correctly, the main motivation for subsetting the SAR data is to avoid the use of highly-correlated neighbouring data points and thus overfitting, wouldn't it be more effective to do the thinning in a more systematic way by omitting more points in data-rich regions rather than subselecting completely randomly without taking data density into account?**

The spatial distribution of the number of SAR observations is influenced by the orbit of the satellites and by the sea-ice extent. Therefore, the spatial distribution of the number of observations used for training the algorithms shown in figure 1 c) is influenced by the seasonal cycle of the sea-ice extent. Furthermore, there is a high variability in the spatial coverage of the MOSAICs (see example below), and some regions can be well covered during a particular day while there are not many observations in these regions during the full training period. Nevertheless, the grid points in these regions can be highly correlated, and a sub-sampling can be necessary. The regions with many observations (typically the Central Arctic) are also the regions with the most reliable observations due to a larger number of overpasses. Therefore, reducing the number of grid points used in the Central Arctic could potentially reduce the quality of the observations used as target variables, and having a negative impact on the random forest algorithms. Though we consider this question as very interesting and relevant, we also think that this is a complex question which is out of the scope of our paper (which is a first attempt of using random forests for calibrating sea-ice drift forecasts). Therefore, we have decided to keep the method which consists of randomly selecting the grid point covered by SAR observations.

Example of MOSAIC showing the speed of sea-ice drift on 13/03/2020 from the CMEMS product named SEAICE\_GLO\_SEAICE\_L4\_NRT\_OBSERVATIONS\_011\_006 (MOSAIC version 2.0).

**P5L130: By evaluating only over the period June-November 2020, doesn't this potentially introduce a seasonal bias for the evaluation? (This also raises the question whether it would be worthwile considering to add the time of the year as an additional predictand?)**

We agree that using the period June-November 2020 for evaluating the forecasts was not ideal, and we have updated the results using the period from June 2020 to May 2021. Furthermore, we have tested using the "day of year" as an additional predictor (see figure below). However, this results in a decrease in forecast accuracy, except for the random forest models predicting the speed of sea-ice drift which are trained using buoy observations. Based on these results, we have decided to discard the "day of year" from the list of predictors. We have added the figure below in the supplementary material and the following sentence in the main paper (section 4.3 Importance of predictor variables):

"Furthermore, we also tested using the day of year as an additional predictor variable (figure S7 of the supplementary material), but adding this variable tends to deteriorate the forecast accuracy for most models, so we decided to discard this variable"

---

## Author Comment (AC3)

We would like to thank the reviewers for their comments which helped us to improve the quality of the manuscript. Please find below our responses to the reviewer's comments.

**Reviewer 2**

###########

Review of Calibration of sea ice drift forecasts using random forest algorithms.

The manuscript describes a new method that post-processes numerical forecasts of sea ice drift using either in situ drifting buoys or satellite images for the training of a random forest algorithm. The results are evaluated against ice drift observations but in a different period, posterior to the training data. The results reveal that there is a systematic component of the ice drift forecast error that can be corrected by machine learning, although the reduction of error remains often less than 10%. The ML algorithms learns more efficiently from the buoys data than from the satellite images, highlighting the problem of temporal averaging.

The drift direction can mostly be improved in the short forecast range, likely because of the unpredictability of wind directions, but interestingly the algorithm is more often able to correct drift speed at longer forecast horizons, which I did not expect. The authors could spice up their article by analysing what their algorithm does to the sea ice drift speed that improves the skills at a 10 days range: are the drifts made systematically faster or slower? This kind of analysis can - if understood - lead to improvements of the forecast systems. More generally, not seeing what the algorithm does to the forecast is a little frustrating. An example of comparison of original to postprocessed and to observed sea ice drifts could be more convincing than cold-blooded skills scores.

We have added the following example of vector maps from TOPAZ4 forecasts and the calibrated forecasts in the supplementary material:

---

## Author Comment (AC4)

We would like to thank the reviewers for their comments which helped us to improve the quality of the manuscript. Please find below our responses to the reviewer's comments.

**Reviewer 3**

**##########**

**Title: Calibration of sea ice drift forecasts using random forest algorithms**
**Authors: Cyril Palerme and Malte Müller**

**This papers present short-term (1-10 days) forecasts of sea ice drifts using a random forest (AI) algorithm and a comparison of the AI forecasts with those of the operational ice-ocean prediction system TOPAZ. The models were trained using buoy or radarsat-derived sea-ice drifts. Predictors include short-term forecast ice speed and angle, wind speed and angle and ice thickness. Results show that the AI forecasts are more skillful than those of TOPAZ irrespective of the training data set. Furthermore, the model trained using sea ice buoys is more skillful in predicting sea ice drift for all lead-time when compared with the model trained with radarsat ice drifts.**

**The paper addresses an interesting question. The use of AI in sea-ice forecasting is relatively new and for this reason, this is a welcome contribution. The paper however is not well written, the introduction is succinct and does not place the work in the context of previous effectively, the model section is entirely missing and there is relatively little discussion of the pre-processing of the input data and its impact on the forecast skill (a factor that is at least equally important as the AI algorithm in producing a skillful model).**

**I recommend that the paper be accepted for publication after the comments below have been addressed (i.e. not rebutted).**

**Major Points:**
**1- The paper must be substantially edited/restructured.**

**I) The introduction is vague, there is a lot of name-dropping but it does not present an in-depth description of the previous work that is required to fully appreciate the content of the paper. I suggest that the authors review the literature more in-depth and revise the introduction substantially, or add a third co-author that works more closely in the field of sea ice forecasting.**

We have completely changed the introduction in order to add a more in-depth description of previous works. We hope that the new introduction meets the expectations of the reviewer. The new 
[revised manuscript text omitted]

**ii)The use of the present perfect to describe the data is odd. "Satellite sea-ice drift observations ... have been used..." (Line 47, 56, etc.). It sounds as though the authors are speaking of previous work by other authors when they are speaking of their own work being presented. The use of the present tense is much more engaging for the reader, or at least the simple past. I.e. We use (used) satellite sea-ice drifts observations..." . These are two examples; there are many more in the paper.**

We agree with this comment and we have replaced the present perfect by the preterit many times in the revised version of the paper.

**Iii) Line 67: "Section 2.2: Data used for the predictor variable". The model used as a reference for the evaluation of the AI models (i.e. TOPAZ) is included here, yet it is not a predictor variable. The forecasted sea ice thickness, concentration, drift speed and angle are all predictor variables but are not described in this section. Only the 10-m wind speed is discussed.**

This section was initially about the data sets used in this study, and not about the predictor variables. However, we have decided to describe the predictor variables in this section in the revised version of the paper:

*"The list of predictor variables is the same for all the models developed in this study, and can be divided into three different categories. First, some geographical information is used with the Cartesian coordinates of the grid points (x and y in the stereographic projection from the TOPAZ4 system), and the distance of the grid point to the nearest coastline in the TOPAZ4 system. Then, the sea ice concentration from passive microwave observations during the day preceding the forecast start date is also used as predictor variable. The variables from sea ice and wind forecasts during the predicted lead time can be considered as the last category. These variables are the wind direction and speed from*

*ECMWF forecasts, as well as the sea ice concentration, thickness, drift speed and direction from TOPAZ4 forecasts. Furthermore, the sea ice drift and concentration observations, as well as ECMWF wind forecasts, were projected onto the grid used in the TOPAZ4 prediction system using nearest-neighbor interpolation before developing the random forest models."*

**iv) TOPAZ is described only very succinctly. It does not say which sea ice model is used, whether there is an ice thickness distribution included, the grid on which the equation are solved, etc.**

We agree with this comment and we have added a more detailed description of TOPAZ4 in section 2.2 of the revised version of the paper:

*"TOPAZ4 is a coupled ice-ocean model for the North Atlantic and the Arctic which provides 10-day forecasts at a spatial resolution of 12.5 km, as well as a reanalysis (Sakov et al., 2012). It uses the version 2.2 of the Hybrid Coordinate Ocean Model (HYCOM; Bleck, 2002; Chassignet et al., 2006) coupled with a one thickness category sea ice model using an elastic-viscous-plastic rheology (Hunke and Dukowicz, 1997) derived from the version 4.1 of the Community Ice CodE (CICE). The model native grid created using conformal mapping has a spatial resolution between 12 and 16 km in the whole domain. An ensemble Kalman filter is used to assimilate satellite sea ice and oceanic observations such as sea ice concentration and drift, along-track sea-level anomalies, sea-surface temperature, as well as in-situ temperature and salinity profiles. Moreover, TOPAZ4 is forced by ECMWF high-resolution weather forecasts at the ocean surface. While TOPAZ4 forecasts are produced daily, data assimilation is only performed on Thursdays, and only the forecasts starting on Thursdays are stored in the long-term archive. Though the TOPAZ4 system provides forecasts with hourly time steps, the forecasts with daily outputs were used here due to the 24-hour span of SAR observations. Previous studies have reported that the speed of sea ice drift is overestimated in the TOPAZ4 system compared to buoy observations from the IABP (Sakov et al., 2012; Xie et al., 2017)"*

**2- The model is section is entirely missing. A mathematical description of the random forest model must be given because AI is relatively new in the field of short-term sea ice forecasting and more simply for the sake of completeness. The reader should not have to read other papers about random forest in order to fully appreciate the content of the current work.**

We agree that the description of the random forest technique was missing, and we have added a longer description of the random forest technique in the section 3.1 of the revised version of the paper:

*"Random forest algorithms consist of an ensemble of decision trees used for regression or classification tasks (Breiman, 2001). In order to avoid overfitting (meaning that the models learn from noise in the training data), independent decision trees must be developed. The independence of decision trees is ensured by using different subsets of the training data set for developing each decision tree, as well as by randomly selecting a fraction of the predictor variables at each node (the node is then split using the variable maximizing a dissimilarity metric among the selected predictors). Each decision tree is trained with a data set created using the bootstrap method, which consists of randomly selecting samples from the original training data with replacement for creating a new data set of the same size as the original one. This results in using about 63 % of the samples from the original data set for training each decision tree.*

*In this study, random forest models were developed for regression using the Python library Scikit-learn-0.23.2 (Pedregosaet al., 2011), and the mean squared error was used to measure the quality of the splits."*

**3- Some pre-processing was done to the data. E.g. the authors used speed and angle rather than latitudinal and meridional components; two different models for speed and angle were proposed. All these decisions leads to improvements in the forecast. Was there any more pre-processing done to the data to improve skill? What was the improvement in the forecast skill using these pre-processing techniques? A few sentences should be included in the discussion about this in section 4.3. I would call this section "Pre-processing of the data".**

The reason why we develop different models for predicting the speed and direction of sea ice drift is that random forest algorithms can only be used to predict one variable. We hope that the new section describing the random forest technique will help to understand this better.

We have added the following statement at the beginning of the new section 3.3 called "Pre-processing of the data":

*"In order to avoid overfitting, it is better to use predictor variables that are not highly correlated. This is why the speed and direction of sea-ice drift, as well as the wind speed and direction, have been used as predictor variables instead of the eastward and northward components. "*

Furthermore we have added the following paragraph at the end of the same section:

*"We also tested random forest models predicting the sea ice drift along the x and y axes of the TOPAZ4 grid using a different set of predictor variables (figure S12 of the supplementary material). For these models, the northward and eastward components of the ECMWF wind forecasts were used as predictors instead of the wind speed and direction, as well as the sea ice drift along the x and y axes from TOPAZ4 forecasts (which are provided by TOPAZ4 outputs) instead of the sea ice drift speed and direction. The direction and speed of sea ice drift were then calculated using the start and end location of the sea ice for comparing those models with the ones directly predicting the direction and speed of sea ice drift. Relatively similar performances were achieved by these models for predicting the direction of sea ice drift, but these models had significantly worse performances for predicting the speed of sea ice drift (larger mean absolute errors of about 12.2 % and 13.7 % on average for the models trained with buoy and SAR observations, respectively). "*

**4- Line 215: A model trained within the Arctic Ocean proper should not be used to predict sea-ice drift in the land-lock sea ice of the Canadian Arctic Archipelago. This is an entirely different dynamical regime. This results and associated discussion should be removed from the paper and from the abstract. Or at least not given such an important presence.**

We agree with this comment, though random forest algorithms could, in principle, be able to identify regions with particular conditions with the spatial coordinates. Nevertheless, we have decided to exclude the Canadian Arctic Archipelago in the revised version of the paper, and we have added the following statement:

*"The Canadian Arctic Archipelago is excluded from our study due to the different characteristics of sea-ice drift in this region (largely influenced by the presence of narrow channels and landfast ice) compared to the rest of the Arctic. Therefore, no data located in the Canadian Arctic Archipelago were used for training and evaluating the random forest models."*

**Minor Points:**
**Line 13-14: Sea ice conditions in the Arctic do not change increasingly faster because of increase in ice drift speed. Increase sea ice drift speed is one such change associated with arctic climate change, but it is not the cause. The cause is thinning of sea ice associated**

**with warmer air temperature, change in cloud phase and its impact on the radiative fluxes at the surface, increased ocean heat flux that interacts closely with sea ice on the shallow arctic shelves, increased storminess in the Arctic, etc**

We have modified the introduction and this statement has been removed.

**Line 61: Why only use sea ice drift speed lower than 5km per day? The mean speed in the Arctic Ocean is 5km /day or ~5cm/sec. It seems that a large amount of data is being ignored without acknowledging it or without providing a rationale for doing so.**

It seems that there has been a misunderstanding here. In the discussion paper, it was written (lines 59-62):
*"While all buoy observations located in an area with a sea-ice concentration higher than 10 % (in the OSI-SAF product described in the next section) were used for training the random forest algorithms, only the buoys with a speed between 0.5 and 100 km per day, located in an area with a sea-ice concentration higher than 10 %, and further than 50 km from the coastlines were used for verification."*

Therefore, we have never excluded the buoy observations with a speed higher than 5 km / day, but only the buoy observations with a speed lower than 0.5 km / day and higher than 100 km / day. In the revised version of the paper, we have changed the threshold of 0.5 km / day to 0.1 km / day. However, we agree that the fraction of observations excluded by this selection was missing, and we have added the following statement in the revised version of the paper:

*"In order to avoid inaccurate and unrealistic values, only the buoys with a speed between 0.1 and 100 km per day, located in an area with a sea ice concentration higher than 10 %, and further than 50 km from the coastlines were used for verification. While only the buoys with a speed between 0.1 and 100 km per day were used for training the random forest models predicting the direction of sea ice drift, all the buoys with a speed lower than 100 km per day were used for training the models predicting the speed of sea ice drift in order to make them able to predict very low speed. During the period from June 2013 to May 2020, about 4.5 % and 0.1 % of the buoys had a speed lower than 0.1 km per day and higher than 100 km per day, respectively. "*

**Line 63: "...have been projected onto the grid used in the TOPAZ4 system". This is not useful information. What grid is used in TOPAZ4? Tri-polar? Curvi-linear? Cube-sphere? I see now that this has been defined later in the paper on Line 103. The grid must be defined when it is first discussed. Is it a Cartesian grid? Or Lat/Lon?**

We have added the projection in the following sentence: *"The drift vectors from buoy observations were then projected onto the polar stereographic grid used in the TOPAZ4 system."* However, we have described the other information in section 2.2 in the description of the TOPAZ4 prediction system (see our response to major point iv).

**Line 79: Which ocean observations are assimilated?**

We have added this information in the revised version of the paper:
*"An ensemble Kalman filter is used to assimilate satellite sea ice and oceanic observations such as sea ice concentration and drift, along-track sea-level anomalies, sea-surface temperature, as well as in-situ temperature and salinity profiles."*

**Line 86: When did the switch to higher resolution happened?**

We have changed this sentence in the revised version of the paper: *"These forecasts have lead times up to 10 days, and the model's spatial resolution changed from about 16 km to 9 km in March 2016 (https://www.ecmwf.int/en/forecasts/documentation-and-support/changes-ecmwf-model)."*

**Line 95: No new paragraph here. "... where R is the Earth's radius, lamda and phi are the..."**

We agree with this comment and we have modified this. The new sentence:
*"where arctan2 represents the 4-quadrant inverse tangent function, R is the Earth's radius, φ and λ represent the latitude and the longitude, and the subscripts "start" and "end" indicate the start and end locations"*

**Equ 4: Unusual notation. arctan(v/u)?**

It is true that "arctan2" was not defined in the text. We have added the following statement in the text:
*"where arctan2 represents the 4-quadrant inverse tangent function"*.

**Line 121: Should it be "data points" instead of "data sets"?**

We agree with this comment, and we have replaced "data sets" by "data points".

**Line 165, Equ. 5: Why Case #3 in Equ. 5? Don't Case #1 and #2 above cover all cases?**

There are also cases where the difference between two directions is between -180 and 180°. However, there was an error in the discussion paper (but not in the analysis). We have corrected this error in the revised version of the paper. When $\Delta D > 180 \Rightarrow$ Error $= \Delta D - 360$ (and not $360 - \Delta D$). Note that this error only affected the direction error, but not the absolute error.

**Line 169-171: This is "Method" material that was already covered earlier. It should be moved to the method section.**

We have removed this statement in the revised version of the paper.

**Line 191. "Moreover the fraction of forecasts improved by the calibration is, on average, larger for the models trained with buoy observations (57.0 %) than for the models trained with SAR observations (54.8 %)". Is this really statistically significant? Errors are provided throughout the paper but it does not transpire in the discussion. The errors should used to assess whether the improvements are significant or not.**

We have added an analysis of the statistical significance using the Wilcoxon signed-rank test, which is suitable for non-parametric data and paired observations (see paragraph below). Note that we have used the Wilcoxon signed-rank test to assess if the difference in absolute errors are significant. The fraction of forecasts improved does not have any statistical distribution, and it is therefore more difficult to assess the statistical significance for this metric.

We have added the following paragraph in the method section:

*"In this study, we used the Wilcoxon signed-rank test to assess the statistical significance of the differences between the absolute errors due to its suitability for non-parametric data (the absolute errors are not normally distributed) and paired observations (the same data set was used for evaluating the different models). We performed this analysis using the two-tailed hypothesis test and the significance level of 0.05."*

And we have describe the statistical significance of the results in the section "4.2 Evaluation of the calibrated forecasts":

[revised manuscript text omitted]

**Line 197: "The fraction of forecast improved is, on average, slightly larger for the models trained with SAR observations (55.3 %) than for the models trained with buoy observations (54.9 %). " Again, is this statistically significant?**

We have answered to this comment in our previous response.

**Line 222: The fraction of data used in the training and validation of the model belongs to the Method section.**

We have moved this statement in the Method section in the revised version of the paper.

**Line 225-230: Repetitive. This was already mentioned in the Method section.**

We have moved this section in the Method section in the revised version of the paper.

**Line 236: Sea ice thickness does not change very much in 10 days. I suspect the ice thickness at t=0 would be equally skillful. This should be mentioned.**

We agree that using sea ice thickness during the initialization of the forecasts should provide a relatively similar information to the algorithms. However, because Pan-Arctic sea ice thickness observations are not available during the summer, it is not possible to use sea ice thickness observations in our random forest models which are used all year round. Therefore, we consider that the best available information is the sea ice thickness forecasts at the predicted lead time, and we do not think that the low temporal variability of sea ice thickness should be mentioned here.

**Section 4.3: The discussion does not present a quantitative assessment of the predictive skill of each predictor. A more quantitative discussion should be provided.**

We agree with this comment, and we have modified this section to present the results more quantitatively. The new section:
*"For both calibration methods, the most important variable for predicting the drift direction is the sea ice drift direction from TOPAZ4 forecasts, followed by the wind direction from ECMWF forecasts (figure 8). On average, the relative importance of sea ice drift direction forecasts is about 1.4 and 1.5 times larger than the one from wind direction forecasts for the models trained with buoy and SAR observations, respectively (figure 8). The sum of the relative importances of these two variables represent, on average, about 46 and 41 % of the sum of all relative importances for the models trained with buoy and SAR observations, respectively. However, the relative importances of these two variables decrease with increasing lead times.*
*Similarly, the sea ice drift speed from TOPAZ4 is the most important variable for predicting the speed of sea ice drift, followed by the wind speed from ECMWF forecasts. On average, the relative importance of sea-ice drift speed forecasts is about 1.7 and 2.2 larger than the one from wind speed forecasts for the models trained with buoy and SAR observations, respectively (figure 8). For the models predicting the speed of sea ice drift, the sum of the relative importances of these two variables represent, on average, about 40 % of the sum of all relative importances for both calibration methods.*

*Furthermore, the relative importances of these two variables also decrease with increasing lead times. On average, the mean absolute errors are reduced by all predictors for the direction and speed of sea ice drift in both calibration methods (figure 9), though some predictor variables do not improve the forecast accuracy for all lead times. While the sea ice concentration observations during the initialization of the forecasts and the sea ice concentration forecasts from TOPAZ4 are correlated, removing one of these variables decreases the accuracy of most random forest models. Therefore, we decided to keep both variables, even if the importances of these variables are probably underestimated due to this correlation. Furthermore, we also tested using the day of year as an additional predictor variable (figure S7 of the supplementary material), but adding this variable tends to deteriorate the forecast accuracy for most models, so we decided to discard this variable.*

*For the models predicting the direction of sea ice drift, removing the drift direction from TOPAZ4 forecasts increases the mean absolute error between 1.1 and 6.7 degrees depending on the lead time and the observations used for the target variable. This is much larger than the differences in mean absolute error when the wind direction from ECMWF forecasts is removed (between 0.1 and 2.2 degrees). For the models predicting the speed of sea ice drift, removing the drift speed from TOPAZ4 forecasts increases the mean absolute error between 0.041 and 0.444 km / day depending on the lead time and the observations used for the target variable. This is also much larger than the differences in mean absolute error when the wind speed from ECMWF forecasts is removed. Surprisingly, removing the wind speed forecasts slightly reduces the mean absolute error (difference of 0.005 km / day) for the model predicting the speed of sea ice drift for a lead time of 4 days trained with SAR observations. For the other models predicting the speed of sea ice drift, removing the wind speed forecasts increases the mean absolute error between 0.001 and 0.127 km / day. Furthermore, the mean absolute errors for the speed of sea ice drift are also considerably reduced by adding the sea ice thickness forecasts from TOPAZ4 (between 0.011 and 0.098 km / day), probably due to the anti-correlation between sea ice thickness and sea ice drift speed (Yu et al., 2020)."*

**Figure 1: Colorbar for the d panel should be changed to avoid saturation.**

The colorbar has been changed.

**Figure 4: Units for sea ice drift should be km/day or ideally cm/sec. It should not be m/day.**

We have changed the unit for sea ice drift speed in the revised version of the paper, and km / day is now used.

**Bruno Tremblay**
**McGill University**